# Synergistic Antimicrobial Effects of Phage vB_AbaSi_W9 and Antibiotics against *Acinetobacter baumannii* Infection

**DOI:** 10.3390/antibiotics13070680

**Published:** 2024-07-22

**Authors:** Yoon-Jung Choi, Shukho Kim, Minsang Shin, Jungmin Kim

**Affiliations:** Department of Microbiology, School of Medicine, Kyungpook National University, Daegu 37224, Republic of Korea; yjchoi8727@knu.ac.kr (Y.-J.C.); shukhokim@knu.ac.kr (S.K.); shinms@knu.ac.kr (M.S.)

**Keywords:** bacteriophage, carbapenem-resistant *Acinetobacter baumannii*, phage–antibiotic synergy, vB_AbaSi_W9

## Abstract

*Acinetobacter baumannii* is a challenging multidrug-resistant pathogen in healthcare. Phage vB_AbaSi_W9 (GenBank: PP146379.1), identified in our previous study, shows lytic activity against 26 (89.66%) of 29 carbapenem-resistant *Acinetobacter baumannii* (CRAB) strains with various sequence types (STs). It is a promising candidate for CRAB treatment; however, its lytic efficiency is insufficient for complete bacterial lysis. Therefore, this study aimed to investigate the clinical utility of the phage vB_AbaSi_W9 by identifying antimicrobial agents that show synergistic effects when combined with it. The *A. baumannii* ATCC17978 strain was used as the host for the phage vB_AbaSi_W9. Adsorption and one-step growth assays of the phage vB_AbaSi_W9 were performed at MOIs of 0.001 and 0.01, respectively. Four clinical strains of CRAB belonging to different sequence types, KBN10P04948 (ST191), LIS2013230 (ST208), KBN10P05982 (ST369), and KBN10P05231 (ST451), were used to investigate phage–antibiotic synergy. Five antibiotics were tested at the following concentration: meropenem (0.25–512 µg/mL); colistin, tigecycline, and rifampicin (0.25–256 µg/mL); and ampicillin/sulbactam (0.25/0.125–512/256 µg/mL). The in vitro synergistic effect of the phage and rifampicin was verified through an in vivo mouse infection model. Phage vB_AbaSi_W9 demonstrated 90% adsorption to host cells in 1 min, a 20 min latent period, and a burst size of 114 PFU/cell. Experiments combining phage vB_AbaSi_W9 with antibiotics demonstrated a pronounced synergistic effect against clinical strains when used with tigecycline and rifampicin. In a mouse model infected with CRAB KBN10P04948 (ST191), the group treated with rifampicin (100 μg/mL) and phage vB_AbaSi_W9 (MOI 1) achieved a 100% survival rate—a significant improvement over the phage-only treatment (8.3% survival rate) or antibiotic-only treatment (25% survival rate) groups. The bacteriophage vB_AbaSi_W9 demonstrated excellent synergy against CRAB strains when combined with tigecycline and rifampicin, suggesting potential candidates for phage–antibiotic combination therapy in treating CRAB infections.

## 1. Introduction

*Acinetobacter (A.) baumannii* is a leading cause of hospital-acquired infections. Its resistance to multiple antibiotics has made treatment particularly challenging [1,2,3,4,5,6,7]. Bacteriophage (phage) therapy has emerged as a potential alternative to antibiotics in combating antibiotic-resistant infections [8,9,10,11,12,13,14,15,16]. Phages are viruses that specifically infect and replicate within bacteria [16,17]. Their unique life cycle involves attaching to the bacterial cell, injecting their genetic material, and utilizing the bacterial machinery to produce new phage particles [18,19]. The benefits of phage therapy include the ability to target specific bacteria while sparing beneficial ones and providing an effective treatment against antibiotic-resistant bacteria [20,21]. Additionally, phages can self-replicate, which allows for high therapeutic effects even in small quantities and at lower production costs, providing economic benefits [19]. Most phages are safe for human use with minimal negative effects on the environment. However, due to their specificity, identifying target bacteria is essential, and there is a risk of diminished therapeutic effects due to immune responses [17]. Furthermore, regulatory and legal barriers may complicate the development and application of phage therapeutics. Issues associated with storage and stability remain challenging and need to be addressed [20,22].

Recent studies have been conducted on combining phages with antibiotics to counteract the limitations of phages, enhance their effect on antibiotic-resistant bacteria, and maximize the effectiveness of antibiotics [23,24]. Phage–antibiotic synergy (PAS) enhances antibacterial efficacy and reduces the likelihood of resistance development against antibiotics and phages [25,26,27]. This synergistic approach leverages the unique mechanisms of action of phages and antibiotics, making it more difficult for bacteria to develop resistance. First, some antibiotics can make the bacterial cell wall more permeable, aiding phage infection. Second, the stress from antibiotic exposure can increase phage replication rates. Antibiotics may induce an SOS response in bacteria—a reaction to DNA damage—making them more susceptible to phage attack. During this response, many phages utilize the activated bacterial machinery to replicate. Furthermore, when phages infect bacteria, they disrupt biofilms, which are protective matrices produced by bacteria that often protect them from antibiotic effects. This disruption exposes the bacteria within, making them more vulnerable to antibiotics. Synergistic action between phages and antibiotics can significantly help combat drug resistance by leveraging the unique mechanisms of each agent. While antibiotics can induce stress responses in bacteria, making them more susceptible to phage attack, phages can disrupt bacterial biofilms, enhancing antibiotic penetration. Recent studies show that PAS therapy has promising potential in treating multidrug-resistant *A. baumannii*. The bacteriophage vB_AbaP_AGC01 demonstrated antibacterial activity against certain *A. baumannii* strains [28]. This antibacterial effect was more pronounced when combined with gentamicin, ciprofloxacin, and meropenem. Phage ΦAB182, when combined with colistin, polymyxin B, ceftazidime, and cefotaxime, was able to eliminate biofilms of MDR *A. baumannii*, with the most effective PAS observed with the combination of ΦAB182 and colistin [27]. In a case where the left tibia of a 42-year-old man was infected with MDR *K. pneumoniae* and MDR *A. baumannii* following a traffic accident, a combination therapy involving antibiotics (colistin and meropenem) and phages (AbKT21phi3 and KpKT21phi1) successfully eliminated the drug-resistant bacteria and rapidly healed the patient’s wound [16,29]. This synergy enhances bactericidal effects against the target bacteria, potentially lowering the antibiotic resistance threshold and reducing the probability of resistance development.

Generally, *A. baumannii* phages exhibit a narrow host range [30,31]. For instance, phage AB1 infects only specific *A. baumannii* strains and does not lyse other clinical isolates [32]. Similarly, the host range of phage vB_Ab4_Hep4 is restricted to particular *A. baumannii* strains, with occasional broader activity due to genetic mutations [33]. In our recent study, eleven phages were isolated that had lytic activity against carbapenem-resistant *A. baumannii* (CRAB). The vB_AbaSi_W9 phage (GenBank: PP146379.1) [34], demonstrated lytic activity against 26 (89.66%) of 29 CRAB strains with various sequence types (STs), exhibiting a broader host spectrum compared to contemporaneously isolated phages such as vB_AbaP_W8 (37.93%, 11/29) and vB_AbaSt_W16 (41.38%, 12/29). Therefore, the extensive host range of vB_AbaSi_W9 suggests its potential as a promising candidate for the treatment of CRAB infections. However, its lytic efficiency was lower than that of other phages. The efficiency of plating (EOP) values of the vB_AbaSi_W9 phage against various clinical isolates of CRAB ranged from 0.001 to 0.009. Additionally, the lysis spots of the vB_AbaSi_W9 phage on the clinical isolates of CRAB were turbid compared with the clear spots on *A. baumannii* ATCC17978. Therefore, this study aimed to investigate the clinical utility of the vB_AbaSi_W9 phage by identifying antimicrobial agents that demonstrate synergistic effects with this phage. The synergistic effect of the vB_AbaSi_W9 phage was investigated in combination with five antibiotics commonly used to treat CRAB infections: meropenem, colistin, ampicillin/sulbactam, tigecycline, and rifampicin. Additionally, the in vivo protective effect of the vB_AbaSi_W9 phage and rifampicin was explored using a mouse model infected with a clinical isolate of CRAB.

## 2. Results

The phage vB_AbaSi_W9 was investigated by determining the optimal MOI, testing adsorption capability, and analyzing the first-stage growth curve. Using *A. baumannii* ATCC17978 as the host bacterium, the optimal MOI for phage vB_AbaSi_W9 ranges from 1 to 1000 (Figure 1A). Adsorption capability testing revealed that phage vB_AbaSi_W9 demonstrated a rapid adsorption rate, with 90% of phages adsorbed within 1 min (Figure 1B). As shown in the one-step growth curve (Figure 1C), the incubation period of vB_AbaSi_W9 was approximately 20 min and the burst size was 117 PFU/cell. 

Subsequent testing of the stability of the phages to temperature and pH revealed that the phages remained stable over a temperature range of 4 °C to 50 °C and a pH range of 4–8 for 2 h (Appendix A). However, at 60 °C, the stability of the phase was significantly reduced by more than log 4 PFU. At 70 °C, the infectivity decreased further, by more than log 6 PFU. Furthermore, at pH values of 3 and 9, the infectivity of the phage decreased by approximately 2 to 3 log PFU in comparison with the neutral condition (pH 6.8) and declined by >5 log PFU at pH values of 2 and 10.

The growth inhibition ability of the phage vB_AbaSi_W9 against various clinical CRAB strains (*A. baumannii* ATCC 17978, KBN10P04948 of ST 191, LIS2013230 of ST 208, KBN10P05982 of ST 369, and ST451 of KBN10P05231) was analyzed using bacterial growth curve analysis (Figure 2). The phage vB_AbaSi_W9 inhibited the growth of all strains for approximately 6 h. After 6 h, a decrease in the growth of *A. baumannii* ATCC17978 (Figure 2A) and KBN10P05982 of ST369 (Figure 2D) was observed compared to that of the control. However, the strains of ST191, ST208, and ST451 (Figure 2B,C,E) did not demonstrate a distinct decrease in growth compared to the control, even at the highest MOI of 1000. 

Table 1 and Figure 3 show the antimicrobial effects of combining phage and antimicrobial agents. When phage vB_AbaSi_W9 was combined with meropenem, the MIC value of meropenem for *A. baumannii* ATCC17978 and ST369 decreased significantly from 256 μg/mL to 0.5 μg/mL (Table 1 and Figure 3A). The FIC value of this combination is 0.002, indicating high synergy. In contrast, the CRAB strains ST191, ST208, and ST451 showed no synergy with the phage combination, as demonstrated by FIC values ranging from 0.5 to 1 (Table 1 and Figure 3A).

When phage vB_AbaSi_W9 was combined with colistin, the MIC of colistin for *A. baumannii* strains ATCC17978, ST208, and ST451 decreased only twofold, resulting in a FIC of 0.5, which indicates no synergy (Table 1 and Figure 3B). However, for the *A. baumannii* ST191 and ST369 strains, the MIC decreased fourfold when treated with phage vB_AbaSi_W9 at MOI 10 and MOI 1000, respectively, resulting in synergy (FIC = 0.250).

When evaluating the combination effect of ampicillin/sulbactam with the phage, *A. baumannii* ATCC17978 showed a substantial decrease in MIC from 128/64 µg/mL to 2/1 µg/mL, with an FIC value of 0.016 indicating high synergy (Table 1 and Figure 3C). However, for the *A. baumannii* ST191 and ST208 strains, the MIC was reduced only twofold, indicating no synergy. In contrast, for the *A. baumannii* ST369 and ST451 strains, the MIC was reduced from 256/128–512/256 μg/mL to 64/32 μg/mL, indicating synergy (FIC < 0.250).

In combining phage vB_AbaSi_W9 and tigecycline, several significant findings were observed (Table 1 and Figure 3D). For *A. baumannii* ATCC17978, the MIC of tigecycline decreased from 4 μg/mL to 0.25 μg/mL (FIC = 0.063). The CRAB strains ST191, ST208, ST369, and ST451 demonstrated a MIC of 8–16 μg/mL with the antibiotic alone. When combined with the phage at MOI 1–1000, the MIC of tigecycline decreased 4–8 times to 0.25–2 μg/mL (FIC < 0.125).

When combining phage vB_AbaSi_W9 and rifampicin, the MIC of *A. baumannii* ATCC17978 decreased from 16 μg/mL to 2 μg/mL (FIC value of 0.125), indicating synergy (Table 1 and Figure 3E). When treated with rifampicin alone, *A. baumannii* strains ST191 and ST451 demonstrated high antibiotic resistance, with MIC values of 256–128 μg/mL. However, when combined with the phage at MOI 0.01–1000, the MIC of rifampicin decreased to 2 μg/mL (FIC = 0.016–0.031). Furthermore, for the CRAB strains ST208 and ST369, the MIC of rifampicin decreased 4–16 times when combined with the phage at MOI 0.001–1000 (FIC = 0.008–0.016). Therefore, for all five tested strains, combining rifampicin and phage vB_AbaSi_W9 led to FIC values below 0.25, indicating a high PAS effect regardless of the clonal type of CRAB strain.

Analysis of the viable cell count (CFU) at the concentrations where synergy was observed (indicated by an asterisk in Figure 3) revealed that combining antibiotics with phage vB_AbaSi_W9 resulted in a reduction in the range of log5 to log8 across all strains and antibiotics tested.

These synergy influences were validated using time–kill curves (Figure 3). The bacterial growth inhibition curves for each concentration of the phage and antibiotics, marked with asterisks, were analyzed. This allowed for a comparison of the antimicrobial effects of individual treatments using either the phage or antibiotics and the combination treatment using the phage and antibiotics over 24 h.

The in vitro synergy influence of phage vB_AbaSi_W9 in combination with rifampicin was further examined using in vivo animal experiments. Figure 4 shows that 100% of the mice survived in the group treated with 100 μg/mL of rifampicin and phage vB_AbaSi_W9 at an MOI of 1. In contrast, the survival rates of the phage-only and rifampicin-only treated groups were 8.3% and 25%, respectively.

## 3. Discussion

Since antibiotics have become less effective due to increasing resistance, the use of phages, either alone or in combination with antibiotics, has emerged as the new solution. This study demonstrated the effectiveness of combining phage and antibiotics as a therapeutic option for CRAB infection.

Generally, phages that target *A. baumannii* have a narrow host range. The phage vB_AbaSi_W9 has a relatively low lytic capacity (EOP, 0.001–0.074) [34]; however, it demonstrates a wide host range, covering CRAB of various clonal types. In this study, the antimicrobial agents that showed synergistic effects when combined with this phage were investigated and it was found that tigecycline and rifampicin demonstrated significant synergistic effects regardless of the clonal type of CRAB. Combining phage vB_AbaSi_W9 and tigecycline resulted in an 8-to-32-fold increase in efficacy, with the MIC decreasing from 4 to 16 μg/mL and 0.25 to 2 μg/mL compared with when the antibiotic was used alone. Similarly, combining phage with rifampicin increased the antimicrobial effectiveness by 8–128 times over rifampicin alone. This study demonstrated that the combination therapy of bacteriophage and antibiotics exhibits a robust synergistic effect against *A. baumannii* infections. Notably, a significant reduction in viable cell counts, ranging from log5 to log8, was observed across all strains and antibiotic combinations. This suggests that combination therapy can reduce bacterial cell counts more effectively than monotherapy.

The interactions between phage vB_AbaSi_W9 and antibiotics such as tigecycline and rifampicin involve complex and multifaceted mechanisms that require detailed analyses for a comprehensive understanding. Phages attach to specific receptors on bacterial cells and inject their genetic material, disrupting bacterial cell walls or defense mechanisms. This disruption improves the ability of antibiotics to penetrate the bacterial cells [30,35,36,37]. Tigecycline inhibits bacterial ribosome function, thereby impeding protein synthesis. Conversely, rifampicin interrupts RNA synthesis, inducing environmental stresses that make the bacteria more vulnerable, potentially enhancing the effectiveness of phage infections [38,39,40]. This stress may weaken their ability to resist page infection. When phage vB_AbaSi_W9 was administered intraperitoneally at a high concentration in the absence of infection, no differences were observed in the behavior, fur texture, hair loss, or crouching posture of the mice compared with the control group, which received SM buffer. The target strain chosen for this purpose was ST191 (KBN10P04948), which demonstrated the highest efficacy among the tested strains. Typically, a neutropenic mouse model induced by cyclophosphamide utilized *A. baumannii* as an opportunistic pathogen. However, administering the *A. baumannii* KBN10P04948 strain at a concentration of 10^9^ CFU without immunosuppression led to a survival rate of 0% within 2 days. This outcome aligns with previous studies indicating that ST191 exhibits a significantly higher virulence potential, consistently surpassing the 60% mortality threshold compared with other STs of *A. baumannii*. Consequently, direct inoculation of *A. baumannii* KBN10P04948 was conducted without immunosuppressive treatment.

Mice inoculated with different concentrations of *A. baumannii* KBN10P04948 showed varying survival rates. Furthermore, at a concentration of 10^10^ CFU, the survival rate was 0% within 12 h post-administration, and at 10^9^ CFU, the survival rate was 0% within 2 days, while lower concentrations showed a 100% survival rate. Consequently, 10^9^ CFU of *A. baumannii* was chosen as the optimal dose for subsequent experiments. Further examination of the combination effect of the phage and rifampicin utilizing an in vivo mouse infection model revealed that 100% of the mice infected with a virulent CRAB clinical strain survived in the group treated with 100 μg/mL of rifampicin and phage vB_AbaSi_W9 at an MOI of 1. Additionally, mice in the group treated with phage vB_AbaSi_W9 and antibiotics demonstrate milder clinical signs (such as ruffled fur, lethargy, and reduced movement) than those in other treatment groups. In vivo experiments demonstrate the potential for clinical therapeutic applications of antibiotic and phage combinations.

While our study demonstrates the potential of phage–antibiotic combinations, several limitations should be addressed in future research. Firstly, our study primarily focused on planktonic bacterial cells. Further investigation is needed to evaluate the efficacy of these combinations against biofilms, which are prevalent in chronic infections [21]. Secondly, the in vivo experiments were conducted in a neutropenic mouse model, which may not fully replicate the complexity of human infections [22]. Additionally, understanding the molecular mechanisms underlying the synergistic interactions will provide insights for optimizing phage therapy [41]. Clinical trials are necessary to validate these findings and establish standardized protocols for phage–antibiotic combination therapies [16]. The potential for phage neutralization in vivo is a significant consideration. Factors such as the host immune response, the presence of antibodies, and the phage’s ability to reach the infection site can influence the effectiveness of phage therapy [16]. While our in vivo experiments demonstrated efficacy, further research is needed to investigate the extent of phage neutralization and strategies to overcome this challenge, such as engineering phages to evade the immune system or using encapsulation techniques to protect phages until they reach their target [16,22].

This study highlights the potential of phage vB_AbaSi_W9 in combination with antibiotics as a promising therapeutic strategy for treating CRAB infections. The significant reductions in viable cell counts observed with combined treatments underscore the value of this approach. Future research should focus on overcoming the limitations identified, including the development of more robust in vivo models and strategies to mitigate phage neutralization [16]. Additionally, exploring the efficacy of these combinations against biofilms and conducting clinical trials will be crucial steps in advancing this therapy towards clinical application [42].

## 4. Materials and Methods

### 4.1. Bacteria

The bacterial strains included in this study were *A. baumannii* ATCC 17978 and four clinical isolates of CRAB with different STs: KBN10P04948 (ST191), LIS2013230 (ST208), KBN10P05982 (ST369), and KBN10P05231 (ST451). These isolates were cultured on Blood Agar Plates (BAP, Synergy innovation, Seoul, Korea) and Brain Heart Infusion Broth and Agar (BHI Broth and BHI Agar, Difco, Detroit, MI, USA). Following the culture, each strain was transferred to BHI Broth, adjusted to a 15% glycerol concentration for preservation, and stored at −70 °C.

### 4.2. Amplification and Purification of Bacteriophage vB_AbaSi_W9

The agar overlay and liquid amplification methods were modified to concentrate and purify the phage vB_AbaSi_W9. *A. baumannii* ATCC 17978 was utilized as the host bacterium for the phage and cultured in BHI broth at 37 °C, shaking at 150 rpm, until the optical density (OD_600_) reached 0.5, corresponding to an actual live bacterial count of approximately 10^8^ CFU/mL [43,44]. This bacterial culture was combined with the purified phage suspension and incubated at 25 °C, shaking at 200 rpm, for 16 h to amplify the phage, after which centrifugation was employed to isolate the phage. The supernatant was then filtered through a 0.22 µm syringe filter, and chloroform was added to constitute 10% of the total volume for further sterilization. To eliminate the remaining media components in the phage suspension and replace them with saline magnesium buffer (SM buffer; 50 mM Tris-HCl, 150 mM NaCl, 10 mM MgCl_2_, 2 mM CaCl_2_, pH 7.5), the “Phage on Tap (PoT)” protocol was used as described by Bonilla et al. [43]. The phage vB_AbaSi_W9 suspension was purified in accordance with the “Phage on Tap” protocol using an Amicon Ultra-15 centrifugal filter (10K, Merck, Dublin, Ireland). Before use, any residual glycerol was removed by centrifuging with sterilized distilled water. The phage suspension was then added to the filter device and centrifuged to separate the filtrate, followed by the addition of SM buffer for a second centrifugation. After repeating this process once, the residual solution in the device was collected, filtered through a 0.25 µm syringe filter, and mixed with 10% chloroform [43]. Finally, glycerol was added to the purified phage suspension to achieve a final concentration of 15%, and the suspension was stored at −25 °C.

### 4.3. Determination of Optimal Multiplicity of Infection (MOI), Phage Adsorption Assay, and Phage One-Step Growth Curve Assay

The *A. baumannii* ATCC 17978 strain, grown to the log phase, was washed thrice with phosphate-buffered saline and adjusted to 10^8^ CFU/mL. Furthermore, to determine the optimal MOI, phages and bacteria were mixed at ratios ranging from 0.001 to 1000 (phages to bacteria) and then incubated for 24 h at 37 °C. After incubation, the phage titer was measured using the double-layer agar method.

The phage adsorption assay was performed as follows. *A. baumannii* ATCC17978 was cultured in 20 mL of BHI medium until it reached the logarithmic phase (concentration: 1 × 10^8^ PFU/mL) at OD_600_. To this culture, 2 mL of phage solution (concentration: 1 × 10^6^ PFU/mL), filtered using a 0.22 μm syringe filter, was added. The final MOI was 0.001, and the mixture was then incubated at 37 °C. Samples of 1 mL were collected at specific intervals (0, 1, 2, 3, 5, 10, and 15 min), centrifuged at 13,500× *g* for 3 min, and filtered through a 0.22 μm filter. Plaque assays were performed with the filtrates to determine the plaque-forming units (PFU), which were calculated by counting 30–300 plaques on the plate after dilution in an SM buffer.

The phage latent period and burst size were determined through first-stage growth analysis [11,35,36,45]. *A. baumannii* ATCC17978 was cultivated in 10 mL of BHI broth until reaching the logarithmic growth phase with an OD_600_ of 0.5. Following centrifugation at 7000× *g* for 15 min, the resulting bacterial pellet was resuspended in 0.9 mL of prewarmed BHI broth. Subsequently, the phage suspension was mixed with the bacterial culture at an MOI of 0.01, followed by incubation at 37 °C for 15 min to facilitate phage adsorption. Following this incubation, the mixture was centrifuged at 13,500× *g* for 3 min to eliminate unattached free phages. The infected bacterial pellet was then transferred to 10 mL of BHI liquid medium, thoroughly mixed, and further cultured at 37 °C. Sampling was performed at 5 min intervals for approximately 60 min during this process. The collected samples were centrifuged at 13,500 rpm for 3 min, after which the supernatant was filtered using a 0.22 μm filter. The phage titer in this filtered supernatant was determined using the double-agar overlays method. Additionally, the burst size was examined by dividing the average PFU/mL during the final three points of the experiment by the mean PFU/mL during the latent phase. The results represent the mean ± standard deviation of three replicates.

### 4.4. Stability of Phage at Different Temperature and pH

To assess thermostability, a 2 mL phage suspension (1.0 × 10^7^ PFU/mL) was incubated for 2 h at 4, 18, 25, 37, 50, 60, and 70 °C. Plaque survival was then determined by analyzing the plaques at each temperature and quantifying the number of surviving plaques. For pH stability testing, 20 mL of phage suspension (1.0 × 10^7^ PFU/mL) was adjusted to pH levels ranging from 2 to 10 using NaOH or HCl. Subsequently, 2 mL of each adjusted pH was transferred to 5 mL sterile tubes and incubated for 2 h at 37 °C. Following incubation, the suspensions were diluted in SM buffer and the surviving phage titer at each pH level was measured through plaque analysis. All tests were performed in triplicate.

### 4.5. Synergy Testing of Phage and Antibiotics

The antibacterial and synergistic effects of phage vB_AbaSi_W9 in combination with antibiotics against CRAB strains were analyzed. The methods outlined in a previous study were modified and the MIC analysis guidelines from the U.S. CLSI were followed [37,38,39,40]. The antibiotics tested included meropenem, colistin, ampicillin/sulbactam, tigecycline, and rifampicin. Four clinical isolates of CRAB belonging to different sequence types (STs) (KBN10P04948 (ST191), LIS2013230 (ST208), KBN10P05982 (ST369), and KBN10P05231 (ST451)) were used to assess the synergy effect of the phage and antibiotic combination. The antibiotics were tested at the following concentration ranges: meropenem (0.25–512 µg/mL); colistin, tigecycline, and rifampicin (0.25–256 µg/mL); and ampicillin/sulbactam (0–0.25/0.125–512/256 µg/mL).

Phage vB_AbaSi_W9 was applied to BAP plates containing *A. baumannii* strains and incubated at 37 °C for 24 h. The strains grown on BAP were then transferred to 10 mL of BHI broth and cultured at 37 °C, shaking at 150 rpm, until the OD_600_ reached 0.5. Subsequently, 200 μL of this culture was added to 5 mL of MHI broth and incubated at 37 °C until reaching a final McFarland standard of 0.5. Following stationary culturing, the culture was diluted 200-fold and a diluted antibiotic solution (20 μL) was added and serially diluted 2-fold. Subsequently, 100 μL of the bacterial culture was added, followed by the phage solution at concentrations ranging from 10^3^ to 10^9^ PFU/mL (MOI 0.001 to 1000). The optical density of each mixture was measured at OD_600_ up to 24 h using a VersaMax microplate reader (Molecular Devices, LLC, San Jose, CA, USA). To calculate the percentage reduction, absorbance readings were adjusted by subtracting the values from negative control wells (untreated). These measurements were averaged after three repetitions to generate synograms.
$$Reduction\;\left(\%\right)=\frac{\left[\left({OD}_{\mathrm{growth}\;\mathrm{control}}-{OD}_{\;\mathrm{treatment}}\right)\right]}{{OD}_{\mathrm{growth}\;\mathrm{control}}}\times 100\;$$

To quantify the synergistic effect between antibiotics and phage against *A. baumannii*, the Fractional Inhibitory Concentration (FIC) was used (45). The FIC quantifies the inhibitory effectiveness of a combination relative to the efficacy of either antibiotics or phage vB_AbaSi_W9 alone. It was determined using the formula below.
$$FIC=\frac{MIC\;of\;the\;drug\;in\;combination}{MIC\;of\;the\;drug\;in\;independent\;}$$

The interpretation of the result is as follows: synergistic, FIC ≤ 0.50; additive, 0.5 < FIC ≤ 1, indifferent, 1< FIC ≤ 2, antagonistic, FIC > 2.

### 4.6. In Vivo Experiment Using Mouse Infection Model

To evaluate the protective effects of the phage vB_AbaSi_W9 and/or rifampicin in vivo, mice were infected with a clinical isolate of CRAB and then treated with the phage and/or rifampicin. The survival of the mice was observed over 7 days. Six-week-old female BALB/c mice were obtained from Orient Bio in Korea and housed for the study. Initially, 12 mice were injected with a 200 µL solution containing a high-concentration 10^14^ PFU of phage vB_AbaSi_W9 to assess phage toxicity. Changes in body weight and survival were monitored. Subsequently, to determine the optimal infectious dose of bacteria, mice were infected intraperitoneally (IP) with 200 µL of *A. baumannii* KBN10P04948 at doses ranging from 10^6^ to 10^10^ CFU. The survival curves of the mice were recorded over a 7-day period, with three mice per dose group. It was observed that the survival rate was 0% on day 2 when mice were infected with 10^9^ CFU of bacteria. Consequently, this concentration was deemed appropriate for in vivo mouse infection. Based on the in vitro findings, the MOI of the phage and concentration of rifampicin were set at 1 and 100 μg/mL, respectively. Phage and/or rifampicin were administered via intraperitoneal injection 30 min after infection with *A. baumannii* KBN10P04948. During the in vivo experiments, clinical signs such as ruffled fur, lethargy, reduced movement, weight loss, and changes in fur texture were monitored and recorded daily. These observations were used to assess the overall health and response of the mice to the treatments. The number of animals used in the in vivo experiments was determined based on power analysis to ensure statistical significance. A minimum of 10 mice per group was chosen to achieve a power of 0.8 with a significance level of *p* < 0.05.

### 4.7. Statistical Analysis

*p*-Values < 0.05 were considered statistically significant. Interaction plots were utilized to assess the potential synergistic effects between the phage and antibiotics through two-way analysis of variance (ANOVA) [46,47]. All statistical analyses were performed using GraphPad Prism 7.04 software (Graph Pad Software, San Diego, CA, USA).

## 5. Conclusions

In this study, the clinical utility of the phage vB_AbaSi_W9 was increased by identifying antimicrobial agents that demonstrate synergistic effects when combined with it. Combining phage vB_AbaSi_W9 with tigecycline or rifampicin demonstrated excellent synergy against clinical strains of CRAB, suggesting the phage’s potential as a valuable resource for phage–antibiotic combination therapy in treating CRAB infections. Furthermore, combining phage vB_AbaSi_W9 and rifampicin provided complete protection in a mouse model infected with a highly virulent clinical strain of CRAB.

## Figures and Tables

**Figure 1 antibiotics-13-00680-f001:**
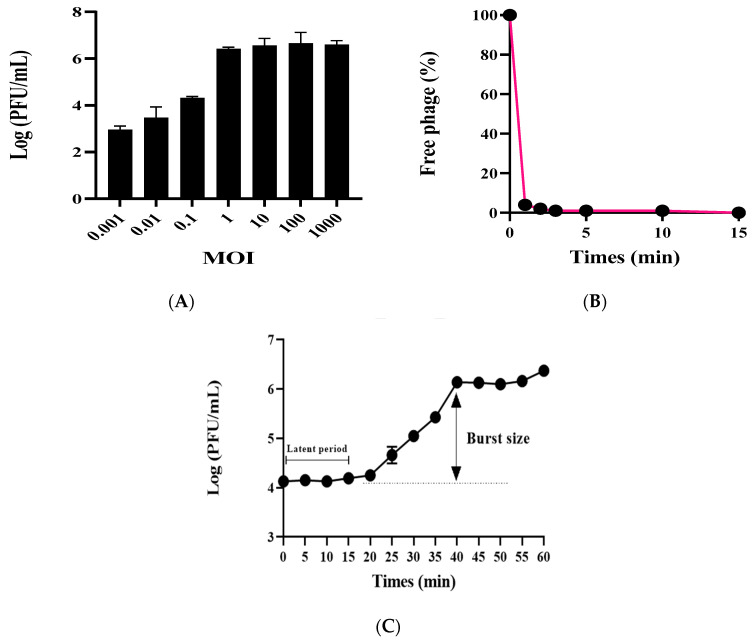
Optimal MOIs, adsorption assay, and one-step growth curve of phage vB_AbaSi_W9. Each data point represents the mean of three independent experiments. Standard deviations are shown as vertical lines. (**A**) Phage yields at different MOIs. (**B**) Adsorption assay of the phage. (**C**) One-step growth curve of the phage.

**Figure 2 antibiotics-13-00680-f002:**
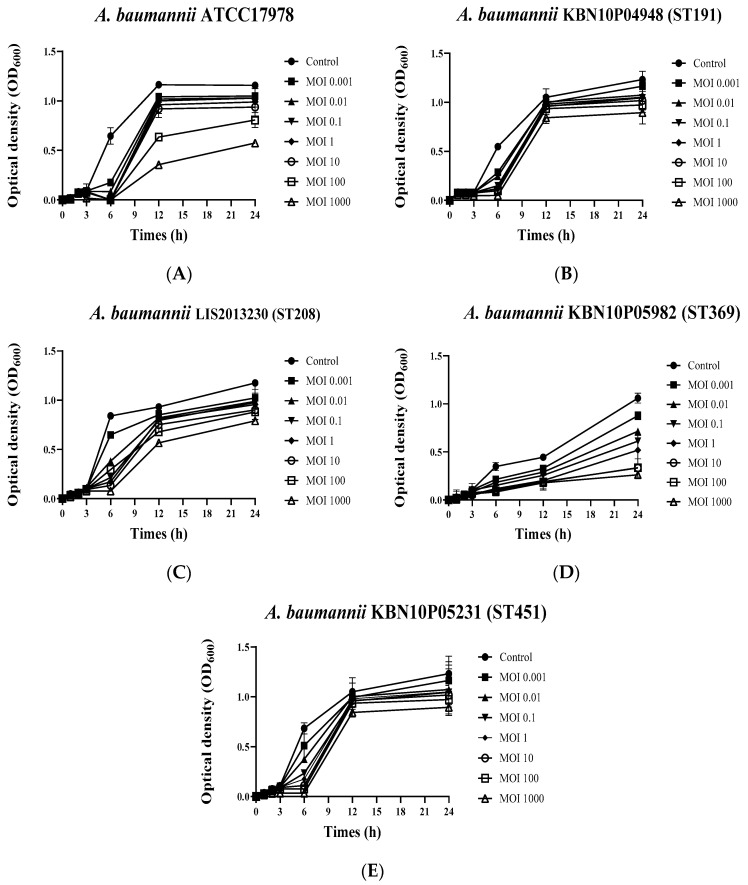
Growth curves of *Acinetobacter baumannii* strains in response to phage vB_AbaSi_W9 at various MOIs. (**A**) *A. baumannii* ATCC 17978, (**B**) *A. baumannii* KBN10P04948 (ST 191), (**C**) *A. baumannii* LIS2013230 (ST 208), (**D**) *A. baumannii* KBN10P05982 (ST 369), and (**E**) *A. baumannii* KBN10P05231(ST451). MOI was confirmed at a ratio of 0.001 to 1000 for each strain. Error bars represent the standard error of the mean (SEM).

**Figure 3 antibiotics-13-00680-f003:**
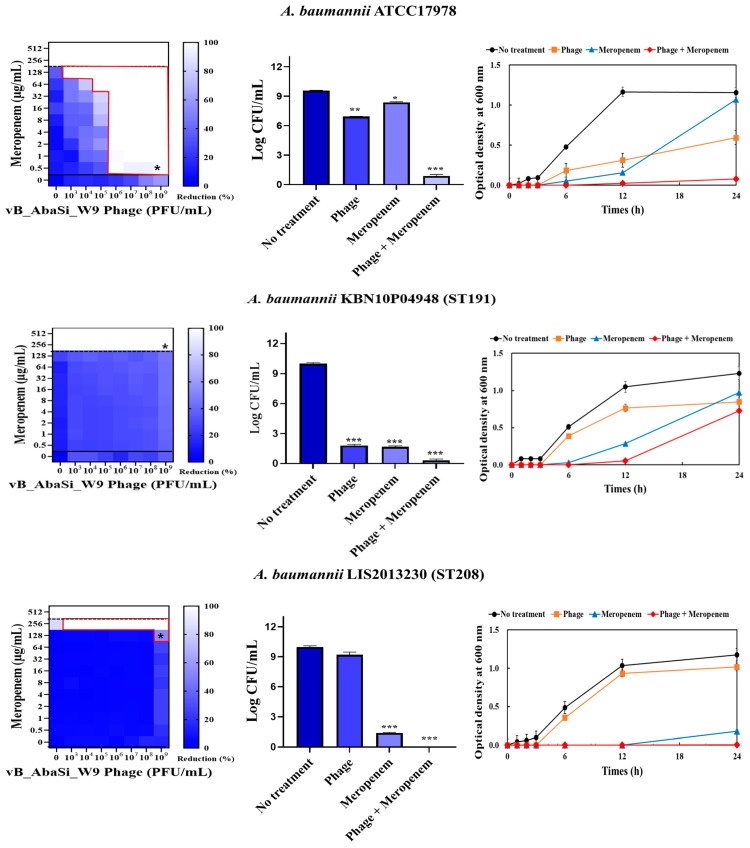
Phage and antibiotic synergy (PAS) tests and time–kill assay. (**A**) Meropenem and phage, (**B**) colistin and phage, (**C**) ampicillin/sulbactam and phage, (**D**) tigecycline and phage, and (**E**) rifampicin and phage. Synograms (t = 24 h) represent the mean reduction percentage of each treatment from three biological replicates: Reduction (%) = [(OD_growth control_ − OD_treatment_) ÷ OD_growth control_] × 100. The regions above the dashed lines indicate antibiotic-mediated killing with highly effective doses. The areas between the solid and dashed lines represent the interaction between the phage and antibiotic regions, while the areas below the solid lines indicate phage-mediated killing with ineffective antibiotic concentrations. Kaplan–Meier survival analysis and the log-rank test were used to determine statistical significance, with thresholds set at red frame (*p* < 0.10), * (*p* < 0.05), ** (*p* < 0.01), and *** (*p* < 0.001) to indicate significant differences compared to the control groups.

**Figure 4 antibiotics-13-00680-f004:**
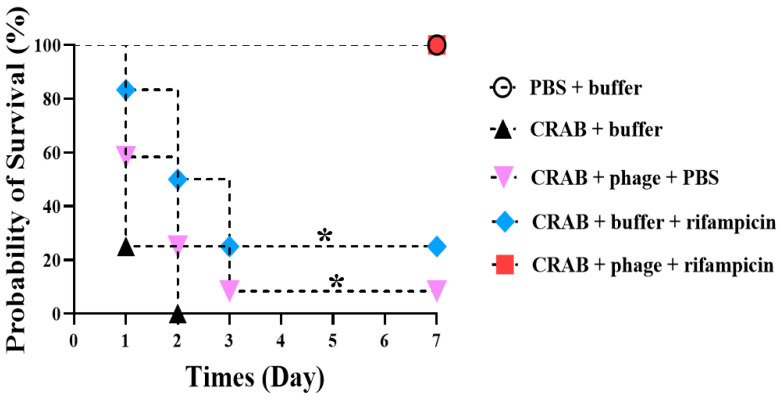
Evaluation of the combined therapeutic efficacy of phage vB_AbaSi_W9 and rifampicin against systemic infection by *Acinetobacter baumannii* KBN10P04948 (ST191) in mice. The bacterial concentration was 10^9^ CFU, the phage was administered at an MOI of 1, the antibiotic concentration was 100 μg/mL, and saline magnesium buffer was used. Statistical significance was determined using Kaplan–Meier survival analysis with the log-rank test, * *p* < 0.05 compared to control groups.

**Table 1 antibiotics-13-00680-t001:** Antimicrobial effects of antibiotics alone or combined with the phage vB_AbaSi_W9 (MOI 1000).

Antimicrobial against	Bacteria	Antimicrobial Susceptibility with Phage			
		Independent ^1^ MIC (µg/mL)	Combined MIC (µg/mL)	^2^ FIC	Effects of ^3^ PAS
Meropenem	*A. baumannii* ATCC17978	256	0.5	0.002	Synergy
	*A. baumannii* KBN10P04948 (ST191)	256	256	1.000	Indifferent
	*A. baumannii* LIS2013230 (ST208)	512	256	0.500	Indifferent
	*A. baumannii* KBN10P05982 (ST369)	256	0.5	0.002	Synergy
	*A. baumannii* KBN10P05231 (ST451)	256	128	0.500	Indifferent
Colistin	*A. baumannii* ATCC17978	4	2	0.500	Indifferent
	*A. baumannii* KBN10P04948 (ST191)	8	2	0.250	Indifferent
	*A. baumannii* LIS2013230 (ST208)	16	8	0.500	Indifferent
	*A. baumannii* KBN10P05982 (ST369)	4	1	0.250	Synergy
	*A. baumannii* KBN10P05231 (ST451)	128	64	0.500	Indifferent
Ampicillin/Sulbactam	*A. baumannii* ATCC17978	128/64	2/1	0.016	Synergy
	*A. baumannii* KBN10P04948 (ST191)	256/128	128/64	0.500	Indifferent
	*A. baumannii* LIS2013230 (ST208)	256/128	128/64	0.500	Indifferent
	*A. baumannii* KBN10P05982 (ST369)	256/128	64/32	0.250	Synergy
	*A. baumannii* KBN10P05231 (ST451)	512/256	64/32	0.004	Synergy
Tigecycline	*A. baumannii* ATCC17978	4	0.25	0.063	Synergy
	*A. baumannii* KBN10P04948 (ST191)	16	2	0.125	Synergy
	*A. baumannii* LIS2013230 (ST208)	8	1	0.125	Synergy
	*A. baumannii* KBN10P05982 (ST369)	16	0.5	0.031	Synergy
	*A. baumannii* KBN10P05231 (ST451)	8	0.25	0.031	Synergy
Rifampicin	*A. baumannii* ATCC17978	16	2	0.125	Synergy
	*A. baumannii* KBN10P04948 (ST191)	256	2	0.008	Synergy
	*A. baumannii* LIS2013230 (ST208)	32	8	0.250	Synergy
	*A. baumannii* KBN10P05982 (ST369)	32	4	0.125	Synergy
	*A. baumannii* KBN10P05231 (ST451)	128	2	0.016	Synergy

^1^ MIC: Minimum Inhibitory Concentration; ^2^ FIC: Fractional Inhibitory Concentration; ^3^ Phage-antibiotic synergy (PAS); FIC values below 0.5 are considered ‘synergistic’, while values above 0.5 are labeled ‘indifferent’.

## Data Availability

Source data supporting the findings of the present study are included in the article and Appendix A.

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
