# Peer review of "Synergistic Antimicrobial Effects of Phage vB_AbaSi_W9 and Antibiotics against Acinetobacter baumannii Infection"

_antibiotics, 2024, doi:10.3390/antibiotics13070680_

Round 1

Reviewer 1 Report

Comments and Suggestions for Authors

I found the article very interesting; the topic about the use of phages has been known for years but the identification of what type and proofing of a synergistic effect with antibiotics to fight against bacterial infections are very stimulating.  The issue of antibiotic resistance requires the use of new tools in clinical practice.

Author Response

Comments 1. 

I found the article very interesting; the topic about the use of phages has been known for years but the identification of what type and proofing of a synergistic effect with antibiotics to fight against bacterial infections are very stimulating. The issue of antibiotic resistance requires the use of new tools in clinical practice. 

Response: We are pleased to hear that you found our article interesting. As you mentioned, while the use of phages has been known for years, identifying specific types and demonstrating their synergistic effects with antibiotics to combat bacterial infections is a highly stimulating area of research. The issue of antibiotic resistance indeed requires the introduction of new tools into clinical practice. In our study, we aimed to demonstrate the potential of combining phages and antibiotics as a treatment option for multidrug-resistant pathogens such as Acinetobacter baumannii. Specifically, we found that the combination of phage vB_AbaSi_W9 with antibiotics like tigecycline and rifampicin showed a significant synergistic effect both in vitro and in vivo. These results hold substantial clinical significance and suggest that further research and clinical applications could contribute significantly to overcoming antibiotic resistance.

Reviewer 2 Report

Comments and Suggestions for Authors

Reviewer’s comments

High rates of antibiotic resistance in the hospitals is a medical crisis nowadays. As an alternative therapeutic option, bacteriophage has been studied and utilized to treat highly resistant bacteria. Phage-antibiotic interaction is one of the important issues in this research area. The study by Choi et al. aims to explore the synergistic effects of phage vB_AbaSi_W9 combined with various antibiotics to enhance its efficacy against CRAB infections. The combination of this phage with tigercycline and rifampicin demonstrated significant synergistic effects against the clinical CRAB strains. A neutropenic mouse model is also used in testing the in vivo effect of the combination of rifampicin and phage vB_AbaSi_W9.

In the background, the author did not mention one of the most important advantages of the phage-antibiotic combination, which is the reduction of the development of resistance of the bacteria to both antibiotics and phages. Some classic literature should also be cited here.

‘Among them, the vB_AbaSi_W9 76 phage (GenBank: PP146379.1, in submission) showed lytic activity against 26 (89.66%) of 77 29 CRAB strains with various sequence types (STs), suggesting it is a promising candidate 78 for treating CRAB infections.’ The author claims that the vB_AbaSi_W9 76 phage has a broader lytic activity against more CRAB strains. That makes a good reason for choosing this phage as a candidate for phage-antibiotic combination treatment. But, can the author mention here briefly how broad or narrow are the lytic activities of common CRAB phage published in other literature? What about the other phages reported by the author recently?

One of the characteristics of CRAB is that they have a strong ability to form biofilm, especially in wound infections. Biofilms are the difficult to be treated only by antibiotics. So, the application of phages is important in this case. When evaluating the effect of a phage-antibiotics combination, it is necessary to evaluate the activity of phage-antibiotics against biofilm formation and mature biofilm of CRAB.

When evaluating the phage killing in figure 2. The authors used optical density for bacterial survival rate, this is very inaccurate. First of all, the bacterial size and aggregation tendency will affect the CFU of bacterial at certain OD. Moreover, when you mention phage killing, you would not calculate the dead bacteria which also counted at certain OD. So, the author should use viable bacterial number instead of using OD value. Making a log10 value would be appreciated.

In Table 1, Antimicrobial effects of antibiotics alone or combined with the phage vB_AbaSi_W9. As the aim of this study is to use the antibiotic together to overcome the disadvantage (low killing ability) of the vB_AbaSi_W9 76 phage with broad activity to CRAB, the authors should show antimicrobial effects of phage alone or combined with the different antibiotics but not what they showed here of antibiotics alone or not.

Line 296, the word culturing in the sentence of ‘Following culturing, each strain was…’ is better to be changed to ‘the culture’.

Comments on the Quality of English Language

Well written. Need minor corrections. 

Author Response

High rates of antibiotic resistance in the hospitals is a medical crisis nowadays. As an alternative therapeutic option, bacteriophage has been studied and utilized to treat highly resistant bacteria. Phage-antibiotic interaction is one of the important issues in this research area. The study by Choi et al. aims to explore the synergistic effects of phage vB_AbaSi_W9 combined with various antibiotics to enhance its efficacy against CRAB infections. The combination of this phage with tigercycline and rifampicin demonstrated significant synergistic effects against the clinical CRAB strains. A neutropenic mouse model is also used in testing the in vivo effect of the combination of rifampicin and phage vB_AbaSi_W9. 

Comments 1. In the background, the author did not mention one of the most important advantages of the phage-antibiotic combination, which is the reduction of the development of resistance of the bacteria to both antibiotics and phages. Some classic literature should also be cited here. 

Response: We have revised the background section to include the advantage of reduced development of resistance when using phage-antibiotic combinations (lines 59-62). Relevant classic literature has also been cited to support this point. 

Comments 2. Among them, the vB_AbaSi_W9 phage (GenBank: PP146379.1, in submission) showed lytic activity against 26 (89.66%) of 29 CRAB strains with various sequence types (STs), suggesting it is a promising candidate 78 for treating CRAB infections.’ The author claims that the vB_AbaSi_W9 phage has a broader lytic activity against more CRAB strains. That makes a good reason for choosing this phage as a candidate for phage-antibiotic combination treatment. But, can the author mention here briefly how broad or narrow are the lytic activities of common CRAB phage published in other literature? What about the other phages reported by the author recently? 

Response: (Lines 84-94) Based on your comments, we have added a comparison of the lytic activities of common CRAB phages published in other literature and a brief mention of other phages we reported recently. To emphasize how the lytic range of vB_AbaSi_W9 compares with other CRAB phages, we included a comparison with phages isolated around the same time, highlighting their host spectrum.  

Comments 3. One of the characteristics of CRAB is that they have a strong ability to form biofilm, especially in wound infections. Biofilms are difficult to be treated only by antibiotics. So, the application of phages is important in this case. When evaluating the effect of a phage-antibiotic combination, it is necessary to evaluate the activity of phage-antibiotics against biofilm formation and mature biofilm of CRAB. 

Response: We appreciate the reviewer's insightful comment regarding the significance of biofilms in CRAB infections. We acknowledge the significance of biofilms in CRAB infections and agree that evaluating the activity of phage-antibiotic combinations against biofilms is important. However, our current study primarily focuses on bacterial cells to establish a foundational understanding of the synergistic effects between phage vB_AbaSi_W9 and antibiotics. Future studies will extend this research to include comprehensive biofilm assays.

Comments 4. When evaluating the phage killing in figure 2, the authors used optical density for bacterial survival rate. This is very inaccurate. First of all, the bacterial size and aggregation tendency will affect the CFU of bacteria at certain OD. Moreover, when you mention phage killing, you would not calculate the dead bacteria which also counted at certain OD. So, the author should use viable bacterial number instead of using OD value. Making a log10 value would be appreciated. 

Response: Thank you for your valuable feedback regarding the use of optical density (OD) measurements. We acknowledge that OD can be influenced by bacterial size and aggregation, which may affect the accuracy of evaluating bacterial survival and phage activity. To address this concern, we have incorporated measurements of viable bacterial counts after 24 hours of incubation. Colony-forming units (CFU) were counted on BHI agar plates to ensure precise evaluation of bacterial viability and phage efficacy (Lines 188-191).

Comments 5. In Table 1, Antimicrobial effects of antibiotics alone or combined with the phage vB_AbaSi_W9. As the aim of this study is to use the antibiotic together to overcome the disadvantage (low killing ability) of the vB_AbaSi_W9 76 phage with broad activity to CRAB, the authors should show antimicrobial effects of phage alone or combined with the different antibiotics but not what they showed here of antibiotics alone or not. 

Response: The phage vB_AbaSi_W9 exhibited low EOP values (0.001~0.009) against various clinical isolates of CRAB, leading us to standardize the MOI to 1000 for all bacterial and phage ratios. Consequently, these values were not included in the table. Therefore, the focus of the table is on comparing the MIC values of antibiotics alone and in combination with the phage to emphasize the enhancement of antimicrobial efficacy. Additionally, the effect of the phage alone has already been demonstrated in Figures 1 and 3. However, in response to the reviewer's suggestion, we have explicitly stated in the table title that an MOI of 1,000 was used for the phage vB_AbaSi_W9.

* Revised legend for Table: Table 1. Antimicrobial effects of antibiotics alone or combined with the phage vB_AbaSi_W9 (MOI 1,000)

Comments 6. Line 296, the word culturing in the sentence of ‘Following culturing, each strain was…’ is better to be changed to ‘the culture’. 

Response: We have made the recommended change to improve the clarity of the text. (Line 297) 

Reviewer 3 Report

Comments and Suggestions for Authors

1.       Authors should introduce, how synergistic action could help in combat drug resistance. One recent article is here to follow. https://pubmed.ncbi.nlm.nih.gov/38136766/

2.       Please include the statistical analysis such as p values description in figure legends along with control details.

3.       Conclusions should be extended including future perspectives.

4.       Please include a separate section in the end describing the limitations of the current study.

Author Response

Comments 1. Authors should introduce, how synergistic action could help in combat drug resistance. One recent article is here to follow. https://pubmed.ncbi.nlm.nih.gov/38136766/

Response 1: (Lines 49-55) We have revised the introduction to include a discussion on how synergistic actions between phages and antibiotics can help combat drug resistance. We have cited the recommended article to support our discussion.

Comments 2. Please include the statistical analysis such as p-values description in figure legends along with control details.

Response: (Lines 414-417) We appreciate this suggestion and have updated the figure legends to include detailed descriptions of the statistical analyses, including p-values and control details.

Figure 2. Growth curves of Acinetobacter baumannii strains in response to phage vB_AbaSi_W9 at various MOIs. (A) A. baumannii ATCC 17978, (B) ST 191 (KNBP04948), (C) ST 208 (LIS2013230), (D) ST 369 (KBN10P05982), and (E) ST451 (KBN10P05231). MOI was confirmed at a ratio of 0.001 to 1000 for each strain. Error bars represent the standard error of the mean (SEM)

Figure 3. Phage and antibiotic synergy (PAS) tests and time-kill assay. (A) meropenem and phage, (B) colistin and phage, (C) ampicillin/sulbactam and phage, (D) tigecycline and phage, (E) rifampicin and phage. Synograms (t = 24 h) represent the mean reduction percentage of each treatment from three biological replicates: Reduction (%) = [(ODgrowth control − ODtreatment) ÷ ODgrowth control] × 100. The regions above the dashed lines indicate antibiotic-mediated killing with highly effective doses. The areas between the solid and dashed lines represent the interaction between the phage and antibiotic regions, while the areas below the solid lines indicate phage-mediated killing with ineffective antibiotic concentrations. Kaplan-Meier survival analysis and log-rank test were used to determine statistical significance, with a p-value of less than 0.05 indicating a significant difference compared to the control groups

Figure 4. Evaluation of the combined therapeutic efficacy of phage vB_AbaSi_W9 and rifampicin against systemic infection by Acinetobacter baumannii KBN10P04948 (ST191) in mice. The bacterial concentration was 109 CFU, the phage was administered at a MOI of 1, the antibiotic concentration was 100 μg/mL, and saline magnesium buffer was used. Statistical significance was determined using Kaplan-Meier survival analysis with log-rank test, **p < 0.05 compared to control groups.

Comments 3. Conclusions should be extended including future perspectives.

Response: (Lines 283-290) We have extended the conclusion to include future perspectives.

Comments 4. Please include a separate section in the end describing the limitations of the current study.

Response: (Lines 267-282) We added a separate section describing the limitations of our study.

Reviewer 4 Report

Comments and Suggestions for Authors

The article reads well and there are no major editorial/language edits.  The experiment conducted were up to scale and the interpretation of results was easy to follow.

A few scientific changes: 

In the in vivo experiments in the materials and methods could indicate what clinical signs were recorded (as stated in the discussion; ruffled fur etc)

Why were the in vivo experiments only done with rifampicin and not with tigecycline?

What would be the interpretation of the optimal dose eg. >50 LD should another strain be used that is not as virulent

Do all the strains grow at the same rate if conditions are similar? ST69 Fig. 3A on meropenem seems to be growing slower in the case of no treatment.

If possible, the figures could be made bigger. 

What is the justification of the number of animals used? could be good to write this in the material and methods section. With what level of significance can we interpret the data especially on the probability of survival and what statistical formula can be used for the number of animals. 

Add to discussion on whether you would think the phages would be neutralized in vivo. 

Author Response

Comments 1. In the in vivo experiments in the materials and methods could indicate what clinical signs were recorded (as stated in the discussion; ruffled fur etc)

Response: (Lines 404-407) We have added details about the clinical signs recorded during the in vivo experiments to the Materials and Methods section.

Comments 2. Why were the in vivo experiments only done with rifampicin and not with tigecycline?

Response: We conducted the in vivo experiments with rifampicin in combination with phage vB_AbaSi_W9 based on its demonstrated superior synergy in vitro. This decision was also influenced by ethical considerations to minimize the number of animals used in the experiments.

Comments 3. What would be the interpretation of the optimal dose eg. >50 LD should another strain be used that is not as virulent

Response: The optimal doses for the in vivo experiments were determined based on the MIC values, irrespective of the strain's virulence. Typically, A. baumannii infections are studied using neutropenic mouse models due to its nature as an opportunistic pathogen. However, for the highly virulent ST191 strain, a dose of 109 CFU was lethal in immunocompetent mice. Despite this, the combination of phage and antibiotics showed significant efficacy. This approach ensured that the selected doses were effective in treating the infection and safe for use in vivo.

Comments 4. Do all the strains grow at the same rate if conditions are similar? ST369 Fig. 3A on meropenem seems to be growing slower in the case of no treatment.

Response: While the growth conditions were standardized, variations in the growth rates among different CRAB strains were observed. For example, ST369 demonstrated slower growth in the absence of treatment compared to other strains. These differences could be attributed to intrinsic growth characteristics and the metabolic diversity of the strains.

Comments 5. If possible, the figures could be made bigger.

Response: We have adjusted the figures in the final version of the manuscript to enhance their visibility and readability.

Comments 6. What is the justification of the number of animals used? could be good to write this in the material and methods section. With what level of significance can we interpret the data especially on the probability of survival and what statistical formula can be used for the number of animals.

Response: (Lines 411-418) The number of animals used in our study was determined based on power analysis to ensure statistical significance. The primary endpoint was the probability of survival, and a significance level (α) of 0.05 was chosen. Using the Kaplan-Meier survival analysis and log-rank test, we calculated the required sample size to detect a significant difference with adequate power (1-β = 0.8). These calculations were guided by previous studies on similar models. I have added this content to lines 382-386.

Comments 7. Add to discussion on whether you would think the phages would be neutralized in vivo.

Response: (Lines 267-279) We have added a discussion on the potential neutralization of phages in vivo.

Round 2

Reviewer 2 Report

Comments and Suggestions for Authors

The authors did not fulfill the requests by Reviewer1. The requests are made to help this manuscript achieve a basic publication quality in the journal Antibiotics. 

Comments on the Quality of English Language

English language is fine. Minor corrections are needed. 

Author Response

We believe we have sincerely addressed the requests made by Reviewer1. However, if there are specific areas where we are lacking, please let us know, and we will make the necessary improvements.

Reviewer 1

Comments 1. In the background, the author did not mention one of the most important advantages of the phage-antibiotic combination, which is the reduction of the development of resistance of the bacteria to both antibiotics and phages. Some classic literature should also be cited here. 

Response : We have revised the background section to include the advantage of reduced development of resistance when using phage-antibiotic combinations (lines 49-57). Relevant classic literature has also been cited to support this point. 

Comments 2. Among them, the vB_AbaSi_W9 phage (GenBank: PP146379.1, in submission) showed lytic activity against 26 (89.66%) of 29 CRAB strains with various sequence types (STs), suggesting it is a promising candidate 78 for treating CRAB infections.’ The author claims that the vB_AbaSi_W9 phage has a broader lytic activity against more CRAB strains. That makes a good reason for choosing this phage as a candidate for phage-antibiotic combination treatment. But, can the author mention here briefly how broad or narrow are the lytic activities of common CRAB phage published in other literature? What about the other phages reported by the author recently? 

Response: (Lines 84-93) Based on your comments, we have added a comparison of the lytic activities of common CRAB phages published in other literature and a brief mention of other phages we reported recently. To emphasize how the lytic range of vB_AbaSi_W9 compares with other CRAB phages, we included a comparison with phages isolated around the same time, highlighting their host spectrum.  

Comments 3. One of the characteristics of CRAB is that they have a strong ability to form biofilm, especially in wound infections. Biofilms are difficult to be treated only by antibiotics. So, the application of phages is important in this case. When evaluating the effect of a phage-antibiotic combination, it is necessary to evaluate the activity of phage-antibiotics against biofilm formation and mature biofilm of CRAB. 

Response: We appreciate the reviewer's insightful comment regarding the significance of biofilms in CRAB infections. We acknowledge the significance of biofilms in CRAB infections and agree that evaluating the activity of phage-antibiotic combinations against biofilms is important. However, our current study primarily focuses on bacterial cells to establish a foundational understanding of the synergistic effects between phage vB_AbaSi_W9 and antibiotics. Future studies will extend this research to include comprehensive biofilm assays.

Comments 4. When evaluating the phage killing in figure 2, the authors used optical density for bacterial survival rate. This is very inaccurate. First of all, the bacterial size and aggregation tendency will affect the CFU of bacteria at certain OD. Moreover, when you mention phage killing, you would not calculate the dead bacteria which also counted at certain OD. So, the author should use viable bacterial number instead of using OD value. Making a log10 value would be appreciated. 

Response: Thank you for your valuable feedback regarding the use of optical density (OD) measurements. We acknowledge that OD can be influenced by bacterial size and aggregation, which may affect the accuracy of evaluating bacterial survival and phage activity. To address this concern, we have incorporated measurements of viable bacterial counts after 24 hours of incubation. Colony-forming units (CFU) were counted on BHI agar plates to ensure precise evaluation of bacterial viability and phage efficacy (Lines 187-190).

Comments 5. In Table 1, Antimicrobial effects of antibiotics alone or combined with the phage vB_AbaSi_W9. As the aim of this study is to use the antibiotic together to overcome the disadvantage (low killing ability) of the vB_AbaSi_W9 76 phage with broad activity to CRAB, the authors should show antimicrobial effects of phage alone or combined with the different antibiotics but not what they showed here of antibiotics alone or not. 

Response: The phage vB_AbaSi_W9 exhibited low EOP values (0.001~0.009) against various clinical isolates of CRAB, leading us to standardize the MOI to 1000 for all bacterial and phage ratios. Consequently, these values were not included in the table. Therefore, the focus of the table is on comparing the MIC values of antibiotics alone and in combination with the phage to emphasize the enhancement of antimicrobial efficacy. Additionally, the effect of the phage alone has already been demonstrated in Figures 1 and 3. However, in response to the reviewer's suggestion, we have explicitly stated in the table title that an MOI of 1,000 was used for the phage vB_AbaSi_W9.

* Revised legend for Table: Table 1. Antimicrobial effects of antibiotics alone or combined with the phage vB_AbaSi_W9 (MOI 1,000)

Comments 6. Line 296, the word culturing in the sentence of ‘Following culturing, each strain was…’ is better to be changed to ‘the culture’. 

Response: We have made the recommended change to improve the clarity of the text. (Line 290)